# Differences in Aggressive Behavior of Individuals with Different Self-Construal Types after Social Exclusion in the Same Cultural Background

**DOI:** 10.3390/bs13080623

**Published:** 2023-07-27

**Authors:** Xiaoli Yang, Yan Zou, Hang Yin, Rui Jiang, Yuan Wang, Fang Wang

**Affiliations:** 1School of Psychology, Northwest Normal University, Lanzhou 730070, China; 2021104101@nwnu.edu.cn (Y.Z.); 2021210536@nwnu.edu.cn (H.Y.); 2021220718@nwnu.edu.cn (R.J.); 2020210529@nwnu.edu.cn (Y.W.); 2Key Laboratory of Behavioral and Mental Health of Gansu Province, School of Psychology, Northwest Normal University, Lanzhou 730070, China; 3Beijing Key Lab of Applied Experimental Psychology, Faculty of Psychology, Beijing Normal University, Beijing 100875, China; fwang@bnu.edu.cn

**Keywords:** social exclusion, self-construal, aggressive behavior, the general aggression model

## Abstract

Aggressive behavior is one of the pervasive and costly negative behaviors in everyday life. Previous studies have shown that individuals who are excluded tend to exhibit more aggressive behaviors, but it is unclear whether the type of self-construction of individuals in the same cultural background will affect the differences in aggressive behavior after being excluded. Therefore, the current study examined the differences in aggressive behavior of individuals with different self-construal types in the same cultural background after social exclusion through two experiments. A total of 128 effective participants were recruited for Experiment 1. Individuals’ self-construal types were classified by the Self-Construal Scale, the cyberball game was used for the manipulation of social exclusion, and the laboratory assistant application paradigm was used to measure individuals’ relational aggression. The results showed that compared with interdependent self-constructors, independent self-constructors exhibited more relational aggression in the exclusion group. A total of 141 effective participants were recruited for Experiment 2. Using the same method as Experiment 1 to classify participants’ self-construal types and induce excluded experiences, the hot sauce paradigm was used to measure individuals’ physical aggression. The results showed that compared with interdependent self-constructors, independent self-constructors exhibited more physical aggression in the exclusion group. The current study helps to understand whether social exclusion negatively impacts individuals with different self-constructors in the same cultural background and provides enlightenment on how individuals who are self-constructors cope with social exclusion.

## 1. Introduction

In China and around the world, aggressive behavior has always been a hot issue of social concern and a focus issue of psychological research [1,2]. Numerous studies have found that aggressive behavior is extremely destructive and can bring high costs to individuals and society [3]. Therefore, to better prevent or intervene in the occurrence of aggressive behavior, it is necessary to examine both external and internal factors that affect aggressive behavior.

Among the most common antecedents of aggressive behavior, social exclusion is strikingly associated with aggressive behavior, and scholars have compared extreme cases such as homicide, school shootings, or gang violence to find that almost all perpetrators have experienced exclusion from their peers [4,5]. Furthermore, laboratory studies have shown that micromanipulation of social exclusion increases aggressive behavior among excluded individuals [6]. However, according to the general aggression model, external factors and internal factors affect an individual’s current state jointly [7]. Studies that tease out personal factors and behavioral tendencies have found that how to respond to others is related to how individuals view their relationship with others, and self-construal is a tendency to define the self based on the relationship between the self and others [8], which has an important impact on individual behavior [9]. Therefore, this study intends to combine extrinsic social exclusion and intrinsic self-construal to explore the impact on aggressive behavior and to further integrate the general aggression model from the perspective of social exclusion and provide enlightenment on how different self-constructors deal with exclusion.

### 1.1. Effects of Social Exclusion on Individual Behavioral Responses

Social exclusion is a phenomenon and process that impedes individuals’ belonging needs and relationship needs when they are ignored or rejected by a social group or others [10,11]. Researchers have found that social exclusion can affect individuals’ basic needs [12], emotions [13], and behavior [14], and the effect of behavior is more complex. That is, scholars have been debating whether individuals will engage in prosocial behavior [15] or aggressive behavior [12] after being excluded. Until recently, a meta-analysis study found that social exclusion does indeed increase the aggression of the excluded [16]. According to the literature review, individuals who are excluded not only choose covert forms of aggression, i.e., a greater tendency to dominate others intentionally [17] and lie and fabricate information [18], but also perform violent forms of aggression, such as relational aggression (i.e., embarrassing others and judging others more negatively) [19,20] and physical aggression (i.e., distributing more chili sauce to the ostracizer, subjecting mice to louder explosions, and stabbing voodoo dolls that symbolized the ostracizer more frequently) [6,12,21].

Regarding why individuals exhibit aggressive behavior after being excluded, the rational choice theory believes that individuals purposefully and rationally choose options that they perceive as beneficial under constrained circumstances [22]. However, distorted perceptions of the potential consequences of one’s actions may lead to wrong or undesirable behavioral decisions. Studies have provided strong empirical evidence to show that even a brief social exclusion episode is sufficient to significantly reduce a person’s perceived meaningful existence [23], and it also leads to individuals’ cognitive deconstruction and self-regulation impairment [24], thereby weakening the sensitivity to the cost of attack and leading to aggressive behavior [25]. The implicit bargain theory also believes that social exclusion breaks an implicit agreement between individuals and society in which individuals would override their urge to break rules as a means of gaining social acceptance. No longer hampered by the necessity to behave appropriately to gain social acceptance, ostracized people lost their willingness to follow rules and thus be more likely to behave aggressively [26]. It is not difficult to see that most of the previous studies on the response of individuals to exclusion have focused on behavioral performance and internal mechanisms, and the exploration of how individual characteristics affect exclusion is not complete. The general aggression model points out that different individuals have different sensitivities to exclusion, which may affect their response to exclusion [7]. Thus, to examine aggressive behavior after being excluded, it is necessary to combine internal personal factors.

### 1.2. Effects of Self-Construal on Individual Behavioral Responses

Self-construal is an individual’s awareness of the relationship between themselves and the world around them and also the way an individual understands the meaning of himself or herself and views his or her relationship with others [9]. Early cross-cultural studies have shown that culture plays an important role in the development of an individual’s self-construal. In general, in Eastern cultures, people consider relationships to be the most important, and even if the costs of building and maintaining relationships outweigh the benefits, they will still gravitate towards relationship maintenance. In Western culture, people emphasize the difference between self and others and pay more attention to the independent self. Given this, scholars regard individuals with cultural characteristics of Eastern collectivism as having interdependent self-construal, while individuals with characteristics of Western individualist culture are regarded as having independent self-construal [27,28].

Early studies have found that when dealing with conflict, interdependent self-constructors are more optimistic about the quality of interpersonal relationships, and are more inclined to actively handle disputes in the case of conflict. Independent self-constructors, on the contrary, care more about self-perception and continue to pervade dissatisfaction with interpersonal conflicts [29]. In addition, as social exclusion is a kind of interpersonal conflict, previous cross-cultural studies found that when Turkish and Indian participant groups are regarded as interdependent self-constructors, and the German and American participant groups are regarded as independent self-constructors, compared with independent self-constructors, interdependent self-constructors report basic needs were higher, and negative emotions and antisocial behavioral intentions were generally lower when after being excluded [30]. However, such studies generally use the individual cultural background as the basis for classifying self-construal types and define antisocial behavior too broadly. Therefore, under the same cultural background, it is necessary to reveal whether there are differences in the self-construal of individuals and whether this difference affects the individual’s aggressive behavior.

### 1.3. The Present Study and Hypotheses

Will individuals in the same cultural context develop different types of self-construal? According to some scholars, with the increasing exchange between countries, individuals from different cultural backgrounds are constantly learning and internalizing multiple cultural systems [31], which means that it is biased to categorize individual self-construal in general terms of cultural background. In addition, recent studies have preliminarily confirmed that individuals in the same cultural background do not have the same type of self-construal, which can affect the early cognitive process of individuals, that is, interdependent self-constructors tend more to cope with negative social cues than independent self-constructors [32]. Therefore, in the same cultural background, we supposed that independent and interdependent self-constructors are affected differently by social exclusion.

In conclusion, based on previous research, the purpose of this study was to examine in the same cultural background whether there are behavioral response differences in different self-constructors from the two dimensions of relational aggression and physical aggression after being excluded. Specifically, Experiment 1 used the laboratory assistant application paradigm to explore the influence of social exclusion on the relational aggression of different self-constructors, and we hypothesized that independent self-constructors would have more relational aggression after being excluded than interdependent self-constructors (H1). Experiment 2 used the hot sauce paradigm to explore the influence of social exclusion on the physical aggression of different self-constructors, and we hypothesized that independent self-constructors would have more physical aggression after being excluded than interdependent self-constructors (H2).

## 2. Experiment 1: Differences in Relational Aggression of Individuals with Different Self-Construal Types after Being Excluded

### 2.1. Methods

#### 2.1.1. Participants

To determine the sample size, we applied a prior power analysis using G* power 3.1 [33]. This analysis revealed that 128 participants were required to reach a good statistical power of 0.80 to detect medium-size (f = 0.25) effects with an alpha value of 0.05 for a 2 × 2 between-subject analysis of variance (ANOVA). One hundred and ninety-three middle school student participants, aged 16–21 years (*M* = 18.54, *SD* = 1.88), were randomly recruited to participate in the screening of self-construal types in a region of Northwest China. Among them, 2 participants were excluded because they scored the same on the independent dimension as the interdependent dimension, leaving 191 participants (interdependent: 129, independent: 62). To balance the number of participants in the two self-construal types, 75 of the 129 interdependent self-construal participants were randomly selected to participate in the follow-up experiment. None of the participants had previously participated in an experiment similar to this one, nor had they experienced any serious negative events during the past week. After excluding nine participants with manipulation check errors, 128 (78 male) participants’ data were entered into the final analysis; the participant information is shown in Table 1.

#### 2.1.2. Design

We used a two-factor between-subject design of 2 (group: acceptance, exclusion) × 2 (self-construal type: independent, interdependent). The group and self-construal type were independent variables, and the participant’s score on relational aggression was the dependent variable.

#### 2.1.3. Materials

(1) The Self-Construal Scale. This scale was devised by Singelis [34]. There are 24 questions in total, including two dimensions of independent (e.g., I am willing to stand out in many ways) and interdependent (e.g., for the benefit of the collective, I will sacrifice my interests), and uses Likert-7 points to score (1 = “not at all”, 7 = “fully conform”). According to previous research, the difference between the average score of all questions in the interdependent dimension and the average score of all questions in the independent dimension is used as the division of individual self-construal, the difference value is positive indicating that the participants are interdependent self-constructors, the difference is negative means that the participants are independent self-constructors, and the difference is zero and does not belong to any group. In previous Chinese participants, the scale has a high degree of reliability and validity [35]. In this experiment, in terms of independent and interdependent dimensions, Cronbach’s alpha coefficients were 0.78 and 0.86, respectively.

(2) Cyberball Game. This paradigm simulates the game scenario of online passing, which is operated by the participant and two other virtual players, and triggers the condition of exclusion or acceptance by controlling the number of catches in the throwing game [36]. Specifically, before the game starts, the participant is told that they need to complete an online passing game to test their mental image; that is, they are asked to throw the ball by clicking on the avatar of either side in the game, and in the process, imagine the gender of the other two players, their appearance, and the possible scenes and inner feelings when placing the online game offline. There were 30 passes in the experiment, and the duration of the experiment was 2–3 min. In the acceptance group, all three players received the ball equally an equal number of times, and the participants in the exclusion group only received two passes after the game started, and then never received the ball again. Subsequently, participants answered to what extent they felt excluded or accepted on Likert-7 (1 = “not at all”, 7 = “extremely”) as a manipulation check of the types of social exclusion.

(3) Measurement of Relational Aggression. We used the “laboratory assistant application paradigm” to measure relational aggression [21]. We told participants that Player 1 on the left side of the screen, who just participated game, was applying to become a laboratory assistant, this position was very competitive, and we would like to hear your opinion, and your evaluation would be one of the important grounds for us to eventually hire this student”. The paradigm consists of 6 items (e.g., the applicant is very unsocial) and uses Likert-7 points to score (1 = “not at all”, 7 = “fully compliant”). In this experiment, the higher scores of participants indicated they have more relational aggression toward the ostracizer. Cronbach’s alpha of the 6 questions was 0.78.

#### 2.1.4. Procedures

Because the number of different self-construal types required in the acceptance and exclusion group should be roughly equal, the previous week, we invited participants to fill out the self-construal scale, and we calculated their self-construal. At the official beginning of the experiment, three master’s students majoring in psychology served as experimenters, and they randomly assigned participants to complete the cyberball game and manipulation check in either the exclusion or acceptance groups according to participants’ self-construal types. Next, the experimenter told participants that they needed to complete the questions in the laboratory assistant application paradigm and fill in demographic information. Finally, when the participants completed all the tasks, the experimenter took the participants to another room and told participants that the experimental tasks they participated in were set in advance by the researcher based on the experimental conditions, hoping not to cause trouble to their lives, and compensated course credits for their participation.

### 2.2. Results

#### 2.2.1. Manipulation Check

Firstly, the accepted feeling reported by the participants was inversely scored, and the average score of the excluded feeling reported by the participants was calculated. An independent sample *t*-test was used to analyze the excluded feeling scores of participants in the acceptance and exclusion groups. The results showed significant differences between the scores of acceptance group participants (*M* = 2.47, *SD* = 0.79) and exclusion group participants (*M* = 5.41, *SD* = 0.82), *t* (126) = 20.56, *p* < 0.001. This showed participants in the exclusion group felt more excluded than those in the acceptance group, which proves that the experimental manipulation was effective.

#### 2.2.2. Examination of Differences in the Influence of Social Exclusion on Relational Aggression in Different Self-Constructors

The descriptive statistics of participants’ scores of relational aggression after exclusion in different self-construal types is shown in Table 2. After controlling for the age and gender of the participants, the data were analyzed using a two-factor ANOVA consisting of 2 (group: acceptance, exclusion) × 2 (self-construal type: independent, interdependent). The main effect of group *F* (1, 122) = 53.94, *p* < 0.001, ηp^2^ = 0.31, and self-construal type *F* (1, 122) = 9.15, *p* = 0.003, ηp^2^ = 0.07, were significant, and the interaction of group and self-construal type was significant *F* (1, 122) = 5.03, *p* = 0.03, ηp^2^ = 0.04 (as shown in Table 3). Further simple effect analysis showed that in the exclusion group, independent self-constructors scored significantly higher than interdependent self-constructors on the relational aggression *F* (1, 122) = 13.99, *p* < 0.001, ηp^2^ = 0.10. But in the acceptance group, independent self-constructors and interdependent self-constructors did not score significantly differently on the relational aggression *F* (1, 122) = 0.29, *p* = 0.59, ηp^2^ = 0.002. In the independent condition, the exclusion group scored significantly higher than the acceptance group on the relational aggression *F* (1, 122) = 42.88, *p* < 0.001, ηp^2^ = 0.26. In the interdependence condition, the exclusion group scored significantly higher than the acceptance group on the relational aggression *F* (1, 122) = 13.42, *p* <0.001, ηp^2^ = 0.10 (as shown in Figure 1). This shows that compared with the acceptance group, individuals in both self-construal types exhibited more relational aggression in the exclusion group. In addition, independent self-constructors produced more relational aggression than interdependent self-constructors in the exclusion group

## 3. Experiment 2: Differences in Physical Aggression of Individuals with Different Self-Construal Types after Being Excluded

### 3.1. Methods

#### 3.1.1. Participants

To determine the sample size, we applied a prior power analysis using G* power 3.1 [33]. This analysis revealed that 128 participants were required to reach a good statistical power of 0.80 to detect medium-size (f = 0.25) effects with an alpha value of 0.05 for a 2 × 2 between-subject analysis of variance (ANOVA). Two hundred and nine middle school student participants, aged 15–20 years (*M* = 17.52, *SD* = 1.22), were randomly recruited to participate in the screening of self-construal types in a region of Northwest China. Among them, 3 participants were excluded because they scored the same on the independent self-construal dimension as the interdependent self-construal dimension, leaving 206 participants (interdependent: 131, independent: 75). To balance the number of participants in the two self-construal types, 75 of the 131 interdependent self-construal participants were randomly selected to participate in the follow-up experiment. None of the participants had previously participated in an experiment similar to this one, nor had they experienced any serious negative events during the past week. After excluding six participants with manipulation check errors, 141 (80 male) participants’ data were entered into the final analysis; the participant information is shown in Table 4.

#### 3.1.2. Design

The design is the same as Experiment 1. The group and self-construal type were independent variables, and the participant’s score on physical aggression was the dependent variable.

#### 3.1.3. Materials

(1) This experiment used the same Self-Construal Scale as Experiment 1, and in terms of independent self-construal and interdependent self-construal dimensions, Cronbach’s alpha coefficients were 0.72 and 0.75, respectively.

(2) The cyberball game was the same as Experiment 1.

(3) Measurement of Physical Aggression. We used a variant of the “hot sauce paradigm” to measure physical aggression [37,38]. Participants were told that “Player 1 on the left side of the screen, who just participated game, is conducting another experiment to measure taste sensitivity. Specifically, you need to choose one cup of the chili sauce that you want Player 1 to taste, out of nine plastic cups containing different grams of chili peppers. Note: Player 1 needs to eat all of the chili sauce you choose for him to complete the experiment. The number of grams of chili sauce in the nine plastic cups was 2, 4, 6, 8, 10, 12, 14, 16, and 18 g. In this experiment, the greater the number of grams of chili selected, the greater the physical aggression toward the ostracizer.

#### 3.1.4. Procedures

The procedure was the same as Experiment 1. Because the number of different self-construal types required in the acceptance and exclusion group should be roughly equal, the previous week, we invited participants to fill out the self-construal scale, and we calculated their self-construal. At the official beginning of the experiment, three master’s students majoring in psychology served as experimenters, and they randomly assigned participants to complete the cyberball game and manipulation check in either the exclusion or acceptance groups according to participants’ self-construal types. Next, the experimenter told the participants that they needed to choose the grams of hot sauce and fill in demographic information. Finally, when the participants completed all the tasks, the experimenter took the participants to another room and told the participants that the experimental tasks they participated in were set in advance by the researcher based on the experimental conditions, hoping not to cause trouble to their lives, and compensated course credits for their participation.

### 3.2. Results

#### 3.2.1. Manipulation Check

Firstly, the accepted feeling reported by the participants was inversely scored, and the average score of the excluded feeling reported by the participants was calculated. An independent sample *t*-test was used to analyze the excluded feeling scores of participants in the acceptance and exclusion groups. The results showed significant differences between the scores of acceptance group participants (*M* = 2.29, *SD* = 0.79) and exclusion group participants (*M* = 5.63, *SD* = 0.73), *t* (139) = 25.91, *p* <0.001. This showed participants in the exclusion group felt more excluded than those in the acceptance group, which proves that the experimental manipulation was effective.

#### 3.2.2. Examination of Differences in the Influence of Social Exclusion on Physical Aggression in Different Self-Constructors

A Likert 9-point scale was used for the number of grams of chili sauce chosen by the participants (e.g., 1 = “2 g”, 9 = “18 g”), and the descriptive statistics of participants’ scores of physical aggression after being excluded for different self-construal types is shown in Table 5. After controlling for the age and gender of the participants, the data were analyzed using a two-factor ANOVA consisting of 2 (group: acceptance, exclusion) × 2 (self-construal type: independent, interdependent). The results showed that there was a significant main effect of the group *F* (1, 135) = 32.74, *p* < 0.001, ηp^2^ = 0.17; there was not a significant main effect of self-construal type was not significant *F* (1, 135) = 1.64, *p* = 0.23, ηp^2^ = 0.01. The interaction of group and self-construal type was significant *F* (1, 135) = 4.23, *p* = 0.04, ηp^2^ = 0.03 (as shown in Table 6). Further simple effect analysis showed that in the exclusion group, independent self-constructors scored significantly higher than interdependent self-constructors on physical aggression *F* (1, 135) = 5.64, *p* = 0.02, ηp^2^ = 0.04. But in the acceptance group, independent self-constructors and interdependent self-constructors did not score significantly differently on the physical aggression *F* (1, 135) = 0.33, *p* = 0.57, ηp^2^ = 0.01. In the independent condition, the exclusion group scored significantly higher than the acceptance group on the physical aggression *F* (1, 135) = 28.73, *p* < 0.001, ηp^2^ = 0.18. In the interdependence condition, the exclusion group scored significantly higher than the acceptance group on physical aggression *F* (1, 135) = 7.03, *p* = 0.01, ηp^2^ = 0.05 (as shown in Figure 2). This shows that compared with the acceptance group, individuals in both self-construal types exhibited more physical aggression in the exclusion group. In addition, independent self-constructors produced more physical aggression than interdependent self-constructors in the exclusion group.

## 4. Discussion

Based on previous studies, the present study adopted the cyberball game to investigate in the same cultural background, the differences in the aggressive behavior of individuals with different self-construal types after being excluded. The results are consistent with previous research trends, and individuals from the same cultural background also exhibit different types of self-construal, which can affect individuals’ responses to social exclusion [39]. Specifically, compared with acceptance conditions, both independent and interdependent self-construal individuals showed more relational aggression and physical aggression in the conditions of exclusion. Moreover, in exclusion conditions, independent self-constructors exhibited more relational aggression and physical aggression than interdependent self-constructors. As a result of these findings, not only will people have a greater understanding of self-construal types, but they will also research into self-construal types, and social exclusion will be broadened. In addition, the present study provides some enlightenment on how individuals with different self-construal types cope with exclusion.

### 4.1. Differences in the Types of Individual Self-Construal in the Same Cultural Background

In previous studies, the concept of self-construal has been derived from the analysis of the relationship between sociocultural factors and self. Scholars generally regard individuals from Western cultural backgrounds as independent self-constructors, and individuals from Eastern cultural backgrounds as interdependent self-constructors [30]. However, with the advent of the globalized society, the rapid development of the economy, and the competition and integration of multiple values, individuals are facing the impact of multiple cultural value orientations, which also affects the formation and development of individual self-construal [40]. This is consistent with the previous study that there are different types of self-construal exhibited in the same cultural context [35]. However, the survey sample also showed that 129 of the 193 respondents in experiment 1 were individuals with interdependent tendencies and 62 were individuals with independent tendencies, and the interdependent and independent tendencies of the 2 were individuals the same. Among the 196 respondents in experiment 2, 131 were individuals with a tendency to depend on each other, 62 were individuals with an independent tendency, and 3 people had the same tendency for dependence and independence. It can be seen that although individuals in the same cultural background may develop two different self-construal types, mainstream culture is still an important factor affecting individual self-construal types.

### 4.2. Differences in Aggressive Behavior among Different Self-Constructors after Being Excluded

According to the general aggression model, different individuals often have different coping strategies when excluded [41]. The current study found that compared with the acceptance group, individuals in both self-construal types exhibited more relational aggression and physical aggression in the exclusion group. In addition, independent self-constructors produced more relational aggression and physical aggression than interdependent self-constructors in the exclusion group. This is consistent with the previous trend that social exclusion appears to lead to more antisocial behavior intentions among independent self-constructors in different cultural contexts [30], and this result can be explained by the following two aspects.

First, the temporal need-threat model believes, different responses to exclusion depend on the type of threat that individuals perceive, while studies have found that independent self-constructors who suffer from social exclusion are more likely to be threatened in terms of efficacy needs and show a desire to reconstruct a sense of control and efficacy, which exacerbates an individual’s tendency to commit aggressive behavior [12]. In addition, previous studies have found that within the same cultural background, interdependent self-constructors exhibited an attentional bias towards smiling faces after being excluded, which is usually an effective indicator of re-acceptance and rebuilding social relationships. On the contrary, independent self-constructors exhibited an attentional bias toward angry faces after being excluded, which is usually a sign of higher aggressive intentions [39]. Therefore, previous research has provided strong evidence for individuals’ behavioral responses after being excluded, that is, interdependent self-constructors may adopt more pro-social behavioral responses to rebuild social relationships, while independent self-constructors may engage in aggressive behavior to avoid being threatened with exclusion again. The current study also confirms this point, that is, compared to interdependent self-constructors, Independent self-constructors exhibit more aggressive behavior after being excluded. Second, an individual’s aggressive behavior results from releasing stress after frustration, according to the frustration-aggressive hypothesis [42], while research finds that interdependent self-constructors value their social groups and are more capable of activating social connections in the face of difficulties. However, independent self-constructors take their uniqueness, agency, self-realization, and sense of control as the foundation, and core of their self-definition [43], they are consequently more likely to experience frustration due to self-definition deviation after being excluded, and then show aggressive behavior.

### 4.3. Differences in Relational Aggression and Physical Aggression among Different Self-Constructors after Being Excluded

In previous studies, independent self-constructors had higher antisocial behavior tendencies after being excluded compared with interdependent self-constructors [30]. However, previous studies did not distinguish antisocial behavior. Our research centered on aggressive behavior, among which, physical aggression that causes physical harm is generally defined as the primary domain, relational aggression that causes psychological harm is defined as the secondary domain, and the primary domain aggressive is more destructive than the secondary domain aggressive [44]. From the results of this study, the effect of relational aggression (*F* (1, 122) = 13.99, *p* < 0.001, ηp^2^ = 0.11) in the exclusion group of independent self-constructors was greater than that of physical aggression *F* (1, 135) = 5.64, *p* = 0.02, ηp^2^ = 0.04. This contradicts the recent meta-analysis studies which have found that it does not differ significantly between primary and secondary domains of aggressive behavior led by social exclusion [16]. The reason may be that this study is to divide different self-construal types to explore individual aggression in the context of Eastern culture, and Eastern culture itself is more implicit in the handling of conflict, so individuals are more likely to adopt less destructive relational aggression than more destructive physical aggression. This suggests that the type of self-construal may also have a moderating effect on the way of aggressive behavior; that is, different self-constructors may perceive the two types of tasks differently, which in turn leads to different degrees of aggressive behavior. Of course, this is only the preliminary conclusion of this study, and future research should verify it.

### 4.4. Implications and Future Prospects

Social interaction mostly occurs between individuals within the same cultural range. Therefore, at the theoretical level, this study provides an empirical basis for refining the relevant theories in the field of social exclusion and also broadens the scope of the self-construal theory. Secondly, at the practical level, the conclusions drawn in this study can provide enlightenment for interpersonal communication of different self-constructors, that is, arrange tasks related to communication and communication for interdependent self-constructors and provide space for personal creation and development of independent self-constructors and to avoid the destructive aggressive behavior of independent self-constructors when communication is poor. However, this study also has shortcomings. First, when selecting participants, this study only measured the self-construal type of the participants and did not activate the self-construal type of the individual, and future studies can further verify the results of this study by activating the paradigm [45] to activate the self-construal type of the participants. Second, the aggression measured in this study was only directed at the ostracizer, and future studies can measure aggression toward the third party. Third, the current study participants only come from a middle school in a certain area of northwest China. Therefore, it only lays the foundation for research in this field. In future experiments, it is necessary to select participant samples from multiple levels and angles to further improve the generalizability of the results.

## 5. Conclusions

Based on previous studies and the general aggression model, the current study explored the differences in aggressive behavior of individuals with different self-construal types after social exclusion through two experiments in the same cultural background. Specifically, we divided the participants’ self-construal types with the Self-Construal Scale, performed manipulation of social exclusion through the cyberball Game, and used the laboratory assistant application paradigm and the hot sauce paradigm to respectively measure individuals’ relational aggression and physical aggression. The results showed that there were two types of self-construal in the same cultural background, independent and interdependent. Compared with interdependent self-constructors, independent self-constructors produced not only more relational aggression after being excluded but also more physical aggression. The current study not only further tests the general aggression model but also provides enlightenment on how individuals who are self-constructors cope with social exclusion.

## Figures and Tables

**Figure 1 behavsci-13-00623-f001:**
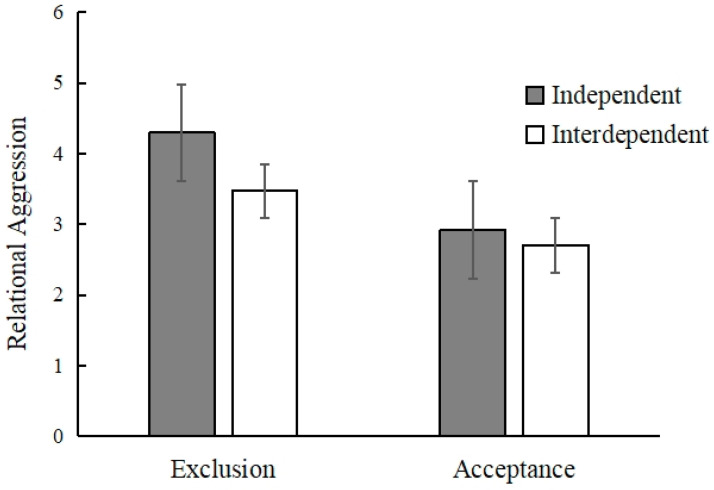
Relational aggression of different self-constructors in the groups of exclusion and acceptance.

**Figure 2 behavsci-13-00623-f002:**
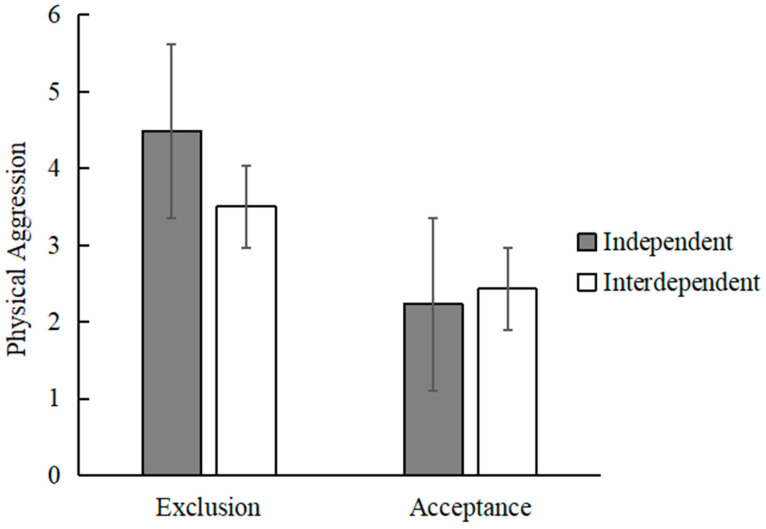
Physical aggression of different self-constructors in the group of exclusion and acceptance.

**Table 1 behavsci-13-00623-t001:** Basic information of participants in Experiment 1.

	Male	Female	Age	Mage	SDage
Exclusion-Independent (*n* = 30)	15	15	16–21	18.09	1.63
Exclusion-Interdependent (*n* = 34)	22	12	16–21	18.05	1.63
Acceptance-Independent (*n* = 31)	24	7	16–21	18.48	1.67
Acceptance-Interdependent (*n* = 33)	17	16	16–21	18.24	1.79
Total (*n* = 128)	78	50	16–21	18.37	1.66

**Table 2 behavsci-13-00623-t002:** Scores of relational aggression among individuals with different self-constructions after being excluded *M*(*SD*).

	Exclusion (*n* = 64)		Acceptance (*n* = 64)	
	Independent (*n* = 30)	Interdependent (*n* = 34)	Independent (*n* = 31)	Interdependent (*n* = 33)
Relational Aggression	4.29 (1.09)	3.47 (0.69)	2.92 (0.80)	2.70 (0.72)

**Table 3 behavsci-13-00623-t003:** The results of ANOVAs conducted on the data collected in Experiment 1.

Effect	Measure	*F* (1, 122)	*p*	ηp^2^
Group	relational aggression	53.94	<0.001	0.31
Self-construal type	relational aggression	9.15	0.003	0.07
Group × Self-construal type	relational aggression	5.03	0.03	0.04

**Table 4 behavsci-13-00623-t004:** Basic information of participants in Experiment 2.

	Male	Female	Age	Mage	SDage
Exclusion-Independent (*n* = 35)	15	20	16–20	17.14	0.88
Exclusion-Interdependent (*n* = 36)	21	15	16–20	17.44	1.27
Acceptance-Independent (*n* = 35)	27	8	15–20	17.60	1.01
Acceptance-Interdependent (*n* = 35)	17	18	16–20	17.60	1.24
Total (*n* = 141)	80	61	15–20	17.45	1.12

**Table 5 behavsci-13-00623-t005:** Scores of physical aggression among individuals with different self-constructions after being excluded *M*(*SD*).

	Exclusion (*n* = 71)		Acceptance (*n* = 70)	
	Independent (*n* = 35)	Interdependent (*n* = 36)	Independent (*n* = 35)	Interdependent (*n* = 35)
Physical Aggression	4.49 (2.06)	3.50 (1.78)	2.23 (1.48)	2.43 (1.07)

**Table 6 behavsci-13-00623-t006:** The results of ANOVAs conducted on the data collected in Experiment 2.

Effect	Measure	*F* (1, 135)	*p*	ηp^2^
Group	physical aggression	32.74	<0.001	0.17
Self-construal type	physical aggression	1.64	0.23	0.01
Group × Self-construal type	physical aggression	4.23	0.04	0.03

## Data Availability

Data are contained within the article and can be made available upon request.

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
