# Peer review of "Differences in Aggressive Behavior of Individuals with Different Self-Construal Types after Social Exclusion in the Same Cultural Background"

_behavsci, 2023, doi:10.3390/bs13080623_

Round 1

Reviewer 1 Report

Thank you for the opportunity to review this article. the topic is interesting and current.

although the material is well presented, logical and coherent, with theoretical argumentation and a current and targeted bibliography, I suggest a series of aspects to consider before the article is published, so that it adds value to both the article and the journal.

so:

- the title does not specify which population it is addressed to

- in the Abstract data should be entered regarding the design of the research: material and methods (subjects, instruments, variables), results with the introduction of some obtained values. in this version, the introduction is too long and the information related to the research very brief.

- in the Introduction chapter, certain details can be introduced as it is too short (perhaps with references to details of theories used, to identified studies, even on the interested population)

- in the Material and methods chapter, clearly presented information is needed regarding the analyzed cultural variables.

- also, in the Results chapter, the values of the obtained data must be presented. in addition, in the current version, some results are presented, but it would be useful to be entered in tables, for ease of reading

- in the Discussions chapter, some paragraphs refer to identified studies but are not found either in the text or in the list of References (see lines 323, 329). also, because the results regarding the analyzed cultural variables are not included, it is more difficult to identify the connection of the study with other studies.

Reviewer 2 Report

The research article title and abstract are appropriate. The study methods are sound and appropriate. The conclusions or summary are accurate and supported by the content. The article is of interest to members of the education research community. 

Author Response

Thanks for the reviewer’s encouragement. Best wishes for you!

Reviewer 3 Report

Provide more demographics collected ion the participants beyond gender and the school. Please describe how you solicited participants and consent. 

Please share some sample laboratory assistant application paradigm questions.

Please describe how you shared the intent of the games.

Please describe how you managed ethical concerns in your research (e.g., safety, confidentiality, privacy, etc.).

Please consider adding information about the researchers in the methods section. I think knowing the professional contexts of the researchers in relationship to the study and its participants gives us readers some important insights in the approach the researchers take to the study.

Discuss your findings in terms of what was previous known and not know about the focus of your research. 

Expand the conclusion.

Did your findings cohere and/or contrast with previous research on similar groups, locations, people, etc.?

Discuss further the limitations of your study. 

Discuss your position on the generalizability of your results.

minor edits needed
